# Demographic and socioeconomic characteristics associated with SARS-CoV-2 reinfection: An observational study

Natasha Nicos Ferreira[1,2☯*], Pedro Manoel Marques Garibaldi[1,2☯],
Gustavo Jardim Volpe[1,2‡], Bruno Belmonte Martinelli Gomes[2], Maira Nilson Benatti[2],
Maria Aparecida Alves Leite Dos Santos Almeida[3], Glenda Renata de Moraes[4],
Dimas Tadeu Covas[5,6], Simone Kashima[6], Rodrigo Tocantins Calado[5,6],
Benedito Antonio Lopes Fonseca[5], Marcos Carvalho Borges[1,2,5‡]

1 Serrana State Hospital, Serrana, Brazil, 2 Clinical Research Center -S, Serrana, Brazil, 3 Butantan Institute, São Paulo, Brazil, 4 Epidemic Service, Serrana, Brazil, 5 University of São Paulo, Ribeirão Preto Medical School, Ribeirão Preto, Brazil, 6 Center for Cell-based Therapy, Blood Center of Ribeirão Preto, Ribeirão Preto, Brazil

☯ These authors contributed equally to this work.
‡ These authors also contributed equally to this work as senior authors.
* natashanicos@heserrana.faepa.br

## Abstract

Since SARS-CoV-2 emergence, risk factors for reinfection have not been totally determined. In this cohort, we analyzed the monthly incidence and outcomes of COVID-19 reinfection and its association with variants of concern, demographic, and socioeconomic factors. An infection case was defined as a positive test for SARS-CoV-2 and reinfection as the presence of a new positive test after 90 days or more of the previous infection. From September 1, 2020, to December 31, 2022, a total of 12,051 cases of COVID-19 were analyzed: 11,129 had one infection, 890 had two infections, and 32 had three infections, yielding a reinfection rate of 7.6%. Female sex was a risk factor for reinfection (RR 1.5, 95% CI 1.23-1.75). A higher risk of reinfection was related to not practicing hand hygiene (RR 1.35, 95% CI 1.14-1.60) and working on-site or from home compared with no work (RR 1.53, 95% CI 1.24-1.87 and RR 3.18, 95% CI 2.02-4.99, respectively). The risk of progressing to moderate or severe disease was higher during the first infection compared with the second (RR 2.12, 95% CI 1.34-3.34). Patients with two or three infections were older than those with one, with a mean age of 75.5 ± 17.3 and 59.6 ± 19.1 years old, respectively (p < 0.002). Our results underscore the importance of targeted public health campaigns and preventive measures for specific demographic groups.

## Introduction

At the end of 2023, the SARS-CoV-2 had already caused more than 700 million infections worldwide, with 6.9 million deaths attributed to COVID-19 [1]. Since the

**Data availability statement:** The data underlying the findings of this study have been deposited in the public repository Zenodo and can be accessed with the following DOI: 10.5281/zenodo.18474588.

**Funding:** The author(s) received no specific funding for this work.

**Competing interests:** The authors have declared that no competing interests exist.

pandemic began, scientific knowledge about this new virus has evolved. However, several points still need to be better elucidated, such as the impact and duration of herd immunity, the protection provided by antibodies produced after infection or vaccination, and the possibility of acquiring the infection more than once.

The first reports of possible reinfections emerged in mid-2020 [2], with an increasing number of cases as the pandemic progressed. New variants emerged despite the high population vaccination rates. The clinical impact of reinfections has been reported with conflicting results. In a previous review, the clinical characteristics and severity of infection were similar between the first and subsequent infections [3]. However, using a national healthcare database of 40,947 reinfections, Bowe *et at*. found that reinfections were associated with additional risks of death, hospitalization, and sequelae during a 6-month follow-up [4]. Additionally, few studies addressed reinfection risk factors [5]. Thus, more studies are still needed to better understand the clinical characteristics, the risk factors, and the dynamic of SARS-CoV-2 reinfections.

In 2020, a surveillance system was implemented in Serrana, SP, Brazil, enhancing access to SARS-CoV-2 testing, patient follow-up, and virus sequencing [6]. Serrana is a mid-sized town with a population of 43,909 in 2022 and socioeconomic characteristics of a commuter city, as many residents work in the neighboring metropolitan area of Ribeirão Preto. This context, combined with its strengthened surveillance system and involvement in a large-scale vaccination trial in 2021, provides a unique setting to study COVID-19 reinfections. This study aims to describe the monthly incidence of COVID-19 reinfections, their clinical characteristics and outcomes, and the association with variants of concern and socioeconomic risk factors.

## Materials and methods

### Ethics statement

This study was conducted in accordance with the ethical standards of the Declaration of Helsinki. The Institutional Ethics Committee of the Medical School of Ribeirão Preto, University of São Paulo (Faculdade de Medicina de Ribeirão Preto – FMRP/USP), approved this analysis as a public health investigation and surveillance and waived the requirement for informed consent (CAAE: 51760221.2.0000.5440). The same ethics approval code was used in a previously published manuscript by the primary author [6], as both studies derived from the same surveillance database but addressed different objectives.

### Study design

This is a cohort conducted in Serrana, located in the state of São Paulo, Brazil. Data were collected from the official database of the Epidemiological Surveillance Department of Serrana and public health surveillance systems (e-SUS and SIVEP-Gripe databases) on January 13, 2023. The authors did not have access to information like name or personal documentation that could identify individual participants, and the data were analyzed in aggregate.

## Study population and period

Serrana is a town with a population of 43,909 inhabitants in 2022, according to an official and compulsory national census [7]. The municipality has a mixed economy, with relevant agricultural activities (mainly sugarcane cultivation), agro-industrial production, commerce, and services. Serrana is considered a commuter town, as a large number of residents work or study in the neighboring city of Ribeirão Preto, a major urban and economic hub located approximately 20 km away. This daily mobility between Serrana and Ribeirão Preto reflects close socioeconomic ties and may have influenced both exposure to SARS-CoV-2 and access to healthcare during the pandemic.

Since September 2020, the Epidemiological Surveillance Department of Serrana has enhanced the surveillance for COVID-19 cases [6]. Any person with one or more possible symptoms of COVID-19, such as coryza, nasal congestion, sore throat, cough, dyspnea, fatigue, fever, muscle pain, headache, nausea, vomiting, diarrhea, dysgeusia, or anosmia, for at least two days could seek medical care at public health units and get tested for free for SARS-CoV-2 with RT-PCR nasal and oral swab. Later, when rapid antigen tests for SARS-CoV-2 became available, few diagnoses were made using this method, although most tests continued to be performed with RT-PCR. This structure enabled the city to be involved in a large stepped-wedge randomized trial to assess CoronaVac's effectiveness in 2021 [8].

Considering the enhanced surveillance program's implementation, the analysis period for this study was from September 1, 2020, to December 31, 2022 (a total of 122 weeks).

## Inclusion criteria

The population of Serrana corresponds to the entire cohort. SARS-CoV-2 infection was considered in individuals with a positive RT-PCR or antigen rapid testing and under active monitoring by the Serrana Epidemiological Surveillance Department. Reinfection was defined by a new positive test result after a minimum period of 90 days from the previous infection [9]. Patients tested within 90 days of a previous COVID-19 diagnosis were excluded from the analysis due to the possibility of persistent infection. Reinfections were identified using the official public health records of the Serrana Epidemiological Surveillance Department. All patients were actively followed for 28 days after each confirmed infection, or until hospital discharge or death. After this period, no further active follow-up was conducted; subsequent reinfections were captured only when individuals returned to the health services and tested positive again. Because diagnostic testing for SARS-CoV-2 was free and widely available through the public health system, the likelihood of capturing the majority of reinfections was considered high.

## Sample analysis and variants of concern

Nasopharyngeal and oropharyngeal swabs collected during this study were processed in Ribeirão Preto Blood Center [6]. Viral RNA was automatically extracted from nasopharyngeal and oropharyngeal swab suspension and the SARS-CoV-2-RT-PCR was executed using the Gene Find- erTM COVID-19 Plus RealAmp kit (OSang Healthcare Co. Ltd., Gyeonggi-do, Korea), which detects fragments of the RdRp, E, and N genes. Genomic sequencing was performed on a sample of 60–80% of all positive samples from each epidemiological week. The period of dominance of each variant, herein called 'waves', was defined based on a positivity rate of at least 50% of the respective variant on an epidemiological week [10]. These definitions were derived from the local epidemiological surveillance system and therefore reflect the variant dynamics in Serrana, which may differ from those described in the international context.

## Data collection

All confirmed cases were followed by the epidemiological surveillance team at the beginning of the symptoms and at days 5, 10, 14, and 28 after symptoms onset. Patients who needed hospitalization were followed up until hospital discharge or death. Disease severity was classified according to the WHO Clinical Progression Scale (WHO-CPS) [11]. Official data inserted in the Brazilian databases was used to identify cases of reinfection [6].

Epidemiological and risk-factor information was obtained through a structured form routinely completed by the epidemiological surveillance nursing staff at the time of SARS-CoV-2 diagnostic testing. This form is part of the official medical record, developed by the local epidemiological surveillance department and reviewed by public health and infectious disease specialists. Data collected included health history, socioeconomic characteristics, and behavioral factors. Education levels were stratified into four groups: 1. No education; 2. Complete or incomplete elementary school; 3. Complete high school or vocational education; 4. Higher education or postgraduate studies. Monthly family income was compared across five strata, considering the Brazilian minimum wage (a minimum wage was R$1,039.00 or USD$182.22): 1. No income; 2. Up to three minimum wages (up to USD$552.66); 3. Three to six minimum wages (USD$552.66 to USD$1,105.32); 4. Six to nine minimum wages (USD$1,105.32 to USD$1,657.98); 5. More than nine minimum wages (Over USD$1.657,98). Preventive measures were also recorded, including use of face masks, social isolation, and more frequent handwashing. Additionally, patients were asked if, in the previous two weeks, they had worked on-site or remotely. Data were collected at each testing event; therefore, individuals who experienced reinfection underwent multiple tests and responded to the same set of questions on each occasion, allowing potential changes in preventive behaviors to be captured over time.

### Statistical analysis

Quantitative variables were summarized using mean, standard deviation (SD), minimum, maximum, median, and interquartile range (IQR, 25th and 75th percentiles), as appropriate. Categorical variables, such as sex and comorbidities, were described using absolute frequencies and percentages.

Incidence rates of reinfection were calculated per month and per variant wave, with corresponding 95% confidence intervals (95% CI). A multivariable Poisson regression model with robust variance was used to estimate the Relative Risk (RR) for demographic and clinical factors associated with reinfection. Given that the overall proportion of reinfections was approximately 7.6% and exceeded 10.0% in some strata, Poisson models with robust variance provide direct and more interpretable RR estimates for cohort data. The multivariable models included age, sex, educational level, monthly family income, household size and number of residents, comorbidities, vaccination status, and behavioral factors such as hand hygiene practices and work status. The same models were applied to compare severity (WHO-CPS $< 4$ vs $\geq 4$) and adherence to preventive measures (such as social isolation, hand hygiene, mask usage, and remote work) between primary infections and reinfections, adjusting for potential confounders identified in univariate analyses of each potential risk factor.

Length of hospitalization between patients with a single infection and those with reinfections was compared using a gamma regression model with random effect, adjusted for confounders. Among patients with WHO-CPS $\geq 4$, Age comparisons between patients with a single infection and those with reinfections were performed using analysis of variance (ANOVA). These models assume the residuals have a normal distribution with mean 0 and constant variance $\sigma^2$. The comparison of sex and comorbidities were performed using the chi-square test.

All analyses were two-sided with a 5% significance level, and 95% Confidence Intervals (CI) were reported. All graphs were generated using R software, version 4.0.4, and the analyses were conducted with SAS 9.4.

## Results

### Description of the study population

During the study period, a total of 36,413 suspected cases of COVID-19 were evaluated, with 12,290 (33.8%) confirmed as positive. Exclusions from the analysis numbered 239 cases, attributed to incomplete data, invalid birthdates, duplicated records, and repeated testing within identical weekly intervals. The final analysis included 12,051 (33.1%) cases of COVID-19. Among these, 11,129 (92.3%) cases corresponded to individuals' first documented infection, 890 (7.4%) to the second documented infection, and 32 (0.3%) to the third documented infection (Fig 1), resulting in a reinfection incidence rate of 7.6%. Most patients were female (57.6%, 66.4%, and 71.9%) with the mean age recorded as 38.4y (SD 18.1), 39.7y (SD 16.5), and 42y (SD 16.6) for one, two, or three infections, respectively. The most prevalent self-reported

PLOS Global Public Health

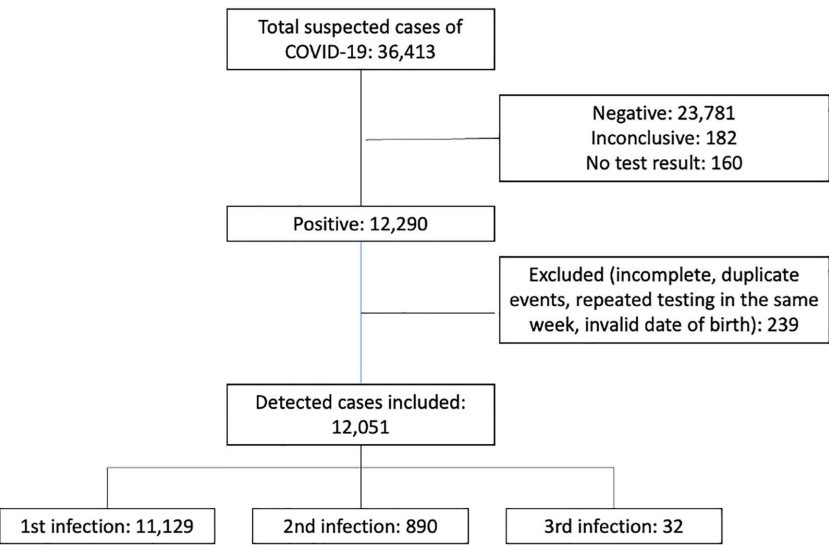

**Fig 1. Comprehensive flow diagram of participant selection and progression in the study.**

comorbid conditions were hypertension (13.2%, 12.7%, and 18.7%) and diabetes (6.2%, 7.4%, and 3.1%). The number of individuals who had already received vaccine doses before each infection progressively increased among the first (20.3%), second (45.5%), and third infections (62.5%). The sociodemographic characteristics of the population showed a higher prevalence of individuals with completed high school education or vocational education and monthly income of up to three minimum wages consistently across all analyzed groups (Table 1).

### Risk factors for SARS-CoV-2 reinfections

For the analysis of reinfection risk factors, only patients with comprehensive data were considered, resulting in 7,361 cases of single infection and 517 cases of second and third infections. The groups of individuals with two and three infections were combined into a single reinfection group due to their relatively small numbers. Female sex was found to be a significant risk factor for reinfection (RR 1.47, 95% CI 1.23-1.75, p < 0.001), whereas age did not exhibit a statistically significant association (RR 1.01, 95% CI 1.00-1.01, p = 0.087). In most of the comparisons among educational levels, individuals with higher levels of education presented a greater risk of having reinfections (high school or vocational education vs elementary school, RR 1.59, 95% CI 1.29-1.96, p < 0,001). Regarding monthly income, having no monthly income appeared to be a risk factor compared to the other income strata (RR 1.82, 95% CI 1.07-3.09, p = 0.028 compared to three to six minimum wages). Furthermore, individuals with a family income of up to three minimum wages presented a lower risk of reinfection compared to those individuals with an income of six to nine minimum wages (RR 0.53, 95% CI 0.36-0.77, p 0.001). House size and number of residents did not show a significant risk of acquiring COVID-19 more than once (Table 2). A comprehensive analysis for all groups can be found in S1 Table.

A lack of hand hygiene practices is positively associated with a higher risk of reinfection (RR 1.35, 95% CI 1.14-1.60, p < 0.001). Employment status also influenced reinfection risks; those working on-site or from home exhibited higher reinfection rates compared to those not working (RR 1.53, 95% CI 1.24-1.87, p < 0.001; and RR 3.18, 95% CI 2.02-4.99, p < 0.001; respectively). Working on-site had a protective effect compared to working from home (RR 0.48, 95% CI 0.31-0.74, p < 0.001) (Table 2). Detailed descriptions of behavioral factors are provided in S3 Table.

**Table 1. Demographic and clinical characteristics of individuals with one, two, or three SARS-CoV-2 infections.**

| | First infection (n = 11129) | Second infection (n = 890) | Third infection (n = 32) |
|---|---|---|---|
| **Age [mean (SD)]** | 38.4 (18.1) | 39.7 (16.5) | 42 (16.6) |
| **Sex (Female)** | 6406 (57.6%) | 591 (66.4%) | 23 (71.9%) |
| **WHO-CPS ≥ 4\*** | 365 (3.3%) | 15 (1.7%) | 1 (3.1%) |
| **Mortality** | 88 (0.8%) | 5 (0.6%) | 0 (0.0%) |
| **Educational background** | | | |
| *No education* | 374 (3.4%) | 11 (1.2%) | 0 (0%) |
| *Elementary school (complete or incomplete)* | 3668 (33.0%) | 184 (20.7%) | 8 (25.0%) |
| *High school (complete) or Vocational education* | 4885 (43.9%) | 416 (46.7%) | 15 (46.9%) |
| *Higher education* | 1449 (13.0%) | 156 (17.5%) | 8 (25.0%) |
| *No information* | 753 (6.8%) | 123 (13.8%) | 1 (3.1%) |
| **Monthly family income\*\*** | | | |
| *No income* | 117 (1.1%) | 17 (1.9%) | 1 (3.1%) |
| *Up to 3 minimum wages (up to $552.66)* | 6341 (57.0%) | 396 (44.5%) | 11 (34.4%) |
| *3 to 6 minimum wages ($552.66 to $1,105.32)* | 2557 (23.0%) | 221 (24.8%) | 7 (21.9%) |
| *6 to 9 minimum wages ($1,105.32 to $1,657.98)* | 281 (2.5%) | 33 (3.7%) | 2 (6.3%) |
| *More than 9 minimum wages (Over $1,657.98)* | 135 (1.2%) | 9 (1.0%) | 0 (0%) |
| *No information* | 1698 (15.3%) | 214 (24.0%) | 11 (34.4%) |
| **House size (number of rooms)** | | | |
| *Up to 3* | 1417 (12.7%) | 101 (11.4%) | 2 (6.3%) |
| *4 or 5* | 5828 (52.4%) | 466 (52.4%) | 14 (43.8%) |
| *6–8* | 2738 (24.6%) | 159 (17.9%) | 7 (21.9%) |
| *9 or more* | 367 (3.3%) | 31 (3.5%) | 2 (6.3%) |
| *No information* | 779 (7.0%) | 133 (14.9%) | 7 (21.9%) |
| **Total house contacts** | | | |
| *0* | 822 (7.4%) | 75 (8.4%) | 4 (12.5%) |
| *1 or 2* | 5186 (46.6%) | 443 (49.8%) | 13 (40.6%) |
| *3 or 4* | 4067 (36.5%) | 309 (34.7%) | 13 (40.6%) |
| *5 or more* | 820 (7.4%) | 42 (4.7%) | 1 (3.1%) |
| *No information* | 234 (2.1%) | 21 (2.4%) | 1 (3.1%) |
| **Comorbidities** | | | |
| *None* | 8695 (78.1%) | 687 (77.2%) | 23 (71.9%) |
| *At least 1* | 2294 (20.6%) | 184 (20.7%) | 8 (25.0%) |
| *Hypertension* | 1469 (13.2%) | 113 (12.7%) | 6 (18.8%) |
| *Diabetes* | 687 (6.2%) | 66 (7.4%) | 1 (3.1%) |
| *Chronic heart disease* | 227 (2.0%) | 13 (1.5%) | 0 (0.0%) |
| *Chronic pulmonary disease* | 279 (2.5%) | 30 (3.4%) | 2 (6.3%) |
| *Chronic kidney disease stage 3–5* | 72 (0.7%) | 1 (0.1%) | 0 (0.0%) |
| **Vaccine doses before infection** | | | |
| *0* | 955 (8.6%) | 5 (0.6%) | 2 (6.3%) |
| *1* | 238 (2.1%) | 3 (0.3%) | 0 (0.0%) |
| *2* | 1602 (14.4%) | 97 (10.9%) | 4 (12.5%) |
| *3* | 2257 (20.3%) | 405 (45.5%) | 20 (62.5%) |
| *No information* | 6077 (54.6%) | 380 (42.7%) | 6 (18.8%) |

*\* WHO-CPS: World Health Organization Clinical Progression Scale*

*\*\* Monthly family income was categorized according to multiples of the Brazilian minimum wage (BRL 1,039.00; approximately USD 182.22 at the time of data collection)*

**Table 2. Risk factors associated with SARS-CoV-2 reinfection.**

| Comparisons | Adjusted relative risk* | 95% Confidence interval | p value |
|---|---|---|---|
| *Age (each 1-year increase)* | 1.01 | 1.00-1.01 | 0.087 |
| *Sex (F vs M)* | 1.47 | 1.23-1.75 | <0.001 |
| *Educational background (Elementary school vs No education)* | 3.56 | 1.15-11.05 | 0.028 |
| *Educational background (High school or Vocational education vs No education)* | 5.67 | 1.83-17.56 | 0.003 |
| *Educational background (Higher education/Postgraduate studies vs No education)* | 5.85 | 1.84-18.55 | 0.003 |
| *Educational background (High school or Vocational education vs Elementary school)* | 1.59 | 1.29-1.96 | <0.001 |
| *Educational background (Higher education/Postgraduate studies vs Elementary school)* | 1.64 | 1.23-2.20 | <0.001 |
| *Family income** (No income vs more than 9 minimum wages)* | 3.43 | 1.25-9.40 | 0.017 |
| *Family income** (6–9 minimum wages vs more than 9 minimum wages)* | 2.78 | 1.12-6.91 | 0.028 |
| *Family income** (up to 3 minimum wages vs 6–9 minimum wages)* | 0.53 | 0.36-0.77 | 0.001 |
| *Family income** (3–6 minimum wages vs 6–9 minimum wages)* | 0.68 | 0.47-0.98 | 0.040 |
| *Family income** (No income vs 3–6 minimum wages)* | 1.82 | 1.07-3.09 | 0.028 |
| *Family income** (up to 3 minimum wages vs 3–6 minimum wages)* | 0.78 | 0.64-0.95 | 0.014 |
| *Family income** (No income vs up to 3 minimum wages)* | 2.33 | 1.39-3.90 | 0.001 |
| *House size (4 or 5 vs 6–8 rooms)* | 1.33 | 1.08-1.65 | 0.008 |
| *Protective measures: Hand hygiene (No vs Yes)* | 1.35 | 1.14-1.60 | <0.001 |
| *Work in the last 2 weeks (Worked from home vs Didn't work)* | 3.18 | 2.02-4.99 | <0.001 |
| *Work in the last 2 weeks (Worked on-site vs Didn't work)* | 1.53 | 1.24-1.87 | <0.001 |
| *Work in the last 2 weeks (Worked on-site vs Worked from home)* | 0.48 | 0.31-0.74 | <0.001 |

*Adjusted relative risks (RRs) were estimated using multivariable Poisson regression with robust variance, adjusted for demographic, socioeconomic, clinical, and behavioral covariates selected a prior.i

** Given the US dollar exchange rate and the Brazilian minimum wage during the study period, one minimum wage corresponded to $184.20. Therefore, the groups were classified into "no income", "up to $552.66", "$552.66 to $1,105.32", "$1,105.32 to $1,657.98" and "more than $1,657.98".

## Severity of SARS-CoV-2 reinfections and viral variants

The first infection posed a higher risk of severe clinical outcomes than the second infection (RR 2.12, 95% CI 1.34-3.34, p<0.001) (S5 Table). Among patients requiring hospitalization due to severe disease, there were no significant differences in comorbidities or sex. However, patients with two or three infections were older than those patients with a single infection, averaging 75.5y (SD 17,3), and 59.6y (SD 19,1), respectively (p<0.002) (Table 3).

## SARS-CoV-2 reinfections across waves with predominance of a single viral variant

In a separate analysis of reinfection dynamics across viral variants waves, regardless of severity, we observed that reinfections occurred at varying rates over time. Reinfections accounted for 20 cases during the Gamma wave (1.0% of 1969 cases), 43 during the Delta wave (2.3% of 1883 cases), 301 during the Omicron wave (7.9% of 3799 cases), and 10 during the Omicron BA.2 sub-lineage wave (13.7% of 73 cases). The remaining cases of reinfection occurred in epidemiological weeks when there was no significant predominance of a single viral variant (Fig 2). Incidence rates of reinfection were then calculated per 10,000 inhabitants in each variant wave. Using this approach, reinfections were estimated at 4.55 (95% CI: 2.56–6.55) per 10,000 inhabitants during the Gamma wave, 9.34 (95% CI: 6.48–12.20) during Delta, 67.41 (95% CI: 59.73–75.09) during Omicron, and 2.28 (95% CI: 0.87–3.69) during Omicron BA.2 sub-lineage. Thus, although Omicron BA.2 showed the highest proportion of reinfections among SARS-CoV-2 cases in that wave, its short circulation period and small number of cases resulted in a lower population-based incidence compared with Delta and Omicron. Overall, Omicron was associated with a markedly higher reinfection burden compared to Gamma, Delta and Omicron

**Table 3. Demographic and clinical characteristics of patients with one or more than one infections and a score ≥4 on the WHO clinical progression scale.**

| Characteristics | One infection (n=330) | Two or three infections (n=15) | p Value* |
|---|---|---|---|
| Age (mean) | 59.6 (19.1) | 75.5 (17.3) | 0.002 |
| Sex (Female) | 151 (45.8%) | 9 (60.0%) | 0.302 |
| *Hypertension* | 107 (32.4%) | 8 (53.3%) | 0.159 |
| *Diabetes* | 67 (20.3%) | 5 (33.3%) | 0.326 |
| *Chronic heart disease* | 48 (14.6%) | 2 (13.3%) | 0.999 |
| *Chronic pulmonary disease* | 16 (4.9%) | 1 (6.7%) | 0.541 |
| *Chronic kidney disease stage 3–5* | 8 (2.4%) | 1 (6.7%) | 0.335 |

*p-values were calculated using Student's t-test for continuous variables (age) and Fisher's exact test for categorical variables.

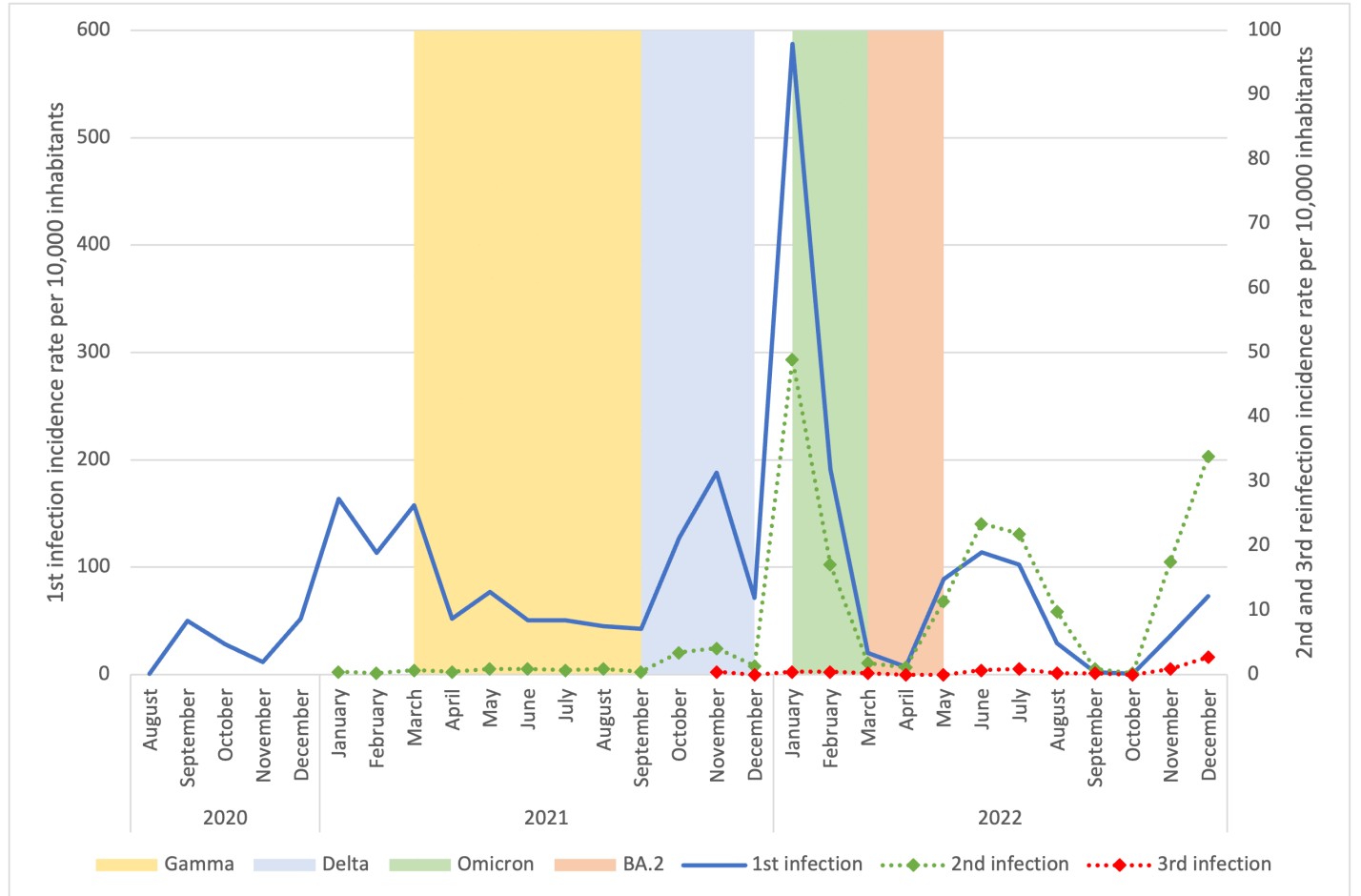

**Fig 2. Monthly reinfection incidence and circulating viral variants.**

BA.2 sub-lineage. Among severe reinfection cases, four occurred during the Gamma wave, five during Omicron, and six during the Omicron BA.1, with no severe reinfections observed during the Delta wave.

## Discussion

Our investigation delineates several factors that heighten the risk for SARS-CoV-2 reinfection. Notably, sex (with a high prevalence in women), higher educational levels, absence of monthly family income, and specific behaviors, such as inadequate hand hygiene practices, emerged as significant predictors of SARS-CoV-2 reinfection. Moreover, the study underscored the influence of different viral variants in the reinfection dynamics. Additionally, patients with reinfections were older and had a less severe disease.

The reinfection rates reported in various studies fluctuate considerably, depending on geographic location, time period, and prevailing viral strains ranging from 0.7% to 28.0% during the Omicron wave [12,13]. In our study, an 7.6% reinfection rate was observed, with notable variation across variant-specific waves: reinfections were rare during Gamma and Delta, but substantially higher during Omicron and Omicron BA.2 sub-lineage. This pattern is consistent with the greater immune escape reported for Omicron lineages, although differences in population immunity and testing practices across epidemic periods may also have contributed. A higher prevalence of reinfection was observed among females. This contrasts with earlier findings from Latin America and Brazil, which did not indicate a significant sex disparity in reinfection rates [14,15]. The implementation of an active surveillance system in Serrana facilitated access to diagnostic testing, regardless of symptom intensity, which contributed to the observed increase in test uptake among women, who generally exhibit higher healthcare-seeking behavior [6]. This could partly explain the sex-specific discrepancy in mild case incidence rates. In addition, behavioral and exposure-related factors may also play a role, as women are more likely to engage in caregiving roles, domestic responsibilities, and occupations with frequent interpersonal contact, which may increase opportunities for SARS-CoV-2 exposure. Similar observations were made in Norway, where women also had an increased risk of reinfection, particularly during the Omicron phase but not during the Alpha and Delta waves, suggesting that differential exposure patterns — rather than biological susceptibility — may underlie this association [16]. These contextual factors may represent potential targets for public health interventions.

Other studies have shown that age, comorbidities, or even lifestyle may be risk factors for acquiring reinfections [17] or severe COVID-19 infections [18]. Our study delved into socioeconomic determinants and factors associated with COVID-19 reinfection and found higher education and lack of monthly family income to be significant risk factors. Individuals with higher education may have a better understanding of COVID-19, clearer symptom interpretation, and a heightened perception of risk, all of which can influence testing behavior. Conversely, individuals without a stable monthly income may experience greater exposure to high-risk settings due to structural and occupational constraints. Determining these risk factors is crucial for tailoring targeted disease control strategies that address unique vulnerabilities or specificities of different demographic groups. Identifying socioeconomic factors contributes to the broader discourse on equitable healthcare and effective pandemic response.

We also demonstrated that working was identified as a risk factor for reinfection, whether done from home or in-site, compared to individuals not working the previous two weeks before diagnosis. As reinfections occurred later in the pandemic, this data might reflect the gradual relaxation of quarantine and social measures and changes in individual behaviors over time [19]. Interestingly, on-site work appeared to afford protection compared to working from home. A possible explanation is that many companies adapted the workplace to minimize contact among employees, whereas isolation at home might be challenging. Since the type of work of the individuals was not addressed in our study, it is also impossible to know whether work-from-home involved face-to-face contact with clients.

The vaccination narrative adds another layer to our understanding of COVID-19 dynamics. As vaccine development began after the onset of the pandemic, and their distribution nationwide in Brazil only started in January 2021, part of the population had their first infection prior to vaccination. Consequently, individuals with a singular infection showed lower

vaccination rates compared to the reinfection groups, illustrating the evolving vaccination landscape over the course of the pandemic [20].

In addition to temporal trends, both prior infection and vaccination may have influenced the clinical course of reinfections. Pre-existing immunity, whether acquired through natural infection, vaccination, or a combination of both, is known to enhance immune memory and mitigate disease severity. In our study, the prevalence of severe cases was higher during the first infection, whereas reinfections were generally associated with milder outcomes, suggesting a protective effect of prior exposure and/or vaccine-induced immunity, in agreement with previous reports [12]

Moreover, patients with severe infection were significantly older in the group of reinfections compared to the group with only one infection. No discernible differences were observed in sex and comorbidity profiles, although analyses comparing severity among individuals with reinfections should be interpreted cautiously due to the small number of severe cases in this group. A previous report from the CDC showed a higher reinfection incidence rate among adults aged 18–49 years (3.0% during Delta and 34.4% during Omicron wave) when compared to adults aged ≥65 years (2.0% during Delta and 18.9% during Omicron wave). This difference can be explained by the higher cumulative incidence of first infections in younger age groups, later eligibility for vaccination, and lower vaccination coverage [12]. This epidemiological scenario did not happen in the municipality of Serrana, as approximately 80% of the adult population was vaccinated within two months, regardless of age group, during the stepped-wedge randomized trial to assess CoronaVac effectiveness in 2021 [8]. Conversely, in another study, younger age groups were associated with a higher risk of reinfection [16].

This study has some limitations. First, different professionals collected data on education and monthly income, and comorbidities were self-reported by the patient. Therefore, some comorbidities may be underestimated. Second, due to the limited sample size, we could not calculate the impact of vaccination doses on reinfection clinical outcomes. Third, the study was conducted in a single town, and findings may not be directly extrapolated to other settings without similar surveillance structures. Fourth, although SARS-CoV-2 diagnostic testing was free and widely available, likely capturing the majority of reinfections, some cases may have gone undetected. Finally, recommendations on preventive measures varied as the pandemic progressed and may have contributed to the reinfection rate.

In conclusion, COVID-19 reinfection was associated with sex, viral variant, educational level, hand hygiene practices, and socioeconomic factors while highlighting age as a determinant for disease severity. These results underscore the importance of targeted public health initiatives and preventive strategies to address the specific needs of diverse demographic groups.

## Supporting information

**S1 Table. Describes epidemiological factors, comorbidities, educational background, socioeconomic data, work-related behavior, and adherence to individual protective measures, comparing individuals with and without reinfection.**
(DOCX)

**S2 Table. Provides a comprehensive univariate and multivariate analysis of all studied factors to assess reinfection risk, using a Poisson regression model with Robust Variance.**
(DOCX)

**S3 Table. Presents the absolute number and percentage of adherence to individual protective measures (face mask usage, social distancing, hand hygiene, remote work) among the groups during the first, second, and third infections.**
(DOCX)

**S4 Table. Provides a complete adjusted prevalence ratio analysis of adherence to preventive measures between the first, second, and third infections, with a 95% confidence interval and p-value.**
(DOCX)

**S5 Table. Presents the adjusted relative risk for severe infections by comparing the first, second, and third infections, using a Poisson regression model with Robust Variance.**
(DOCX)

## Acknowledgments

We would like to express our gratitude to the Municipal Government of Serrana for their invaluable support in providing the data used in this study. Our thanks also go to the Blood Center of the Hospital das Clínicas de Ribeirão Preto for their prompt and efficient execution of RT-PCR tests. We are deeply grateful to the Ethics Committee of the Medical School of Ribeirão Preto for their guidance and approval of the study protocols. Additionally, we thank the Butantan Institute for their support in implementing the enhanced surveillance program.

## Author contributions

**Conceptualization:** Natasha Nicos Ferreira, Pedro Manoel Marques Garibaldi, Gustavo Jardim Volpe, Bruno Belmonte Martinelli Gomes, Maira Nilson Benatti, Dimas Tadeu Covas, Rodrigo Tocantins Calado, Benedito Antonio Lopes Fonseca, Marcos Carvalho Borges.

**Data curation:** Natasha Nicos Ferreira, Pedro Manoel Marques Garibaldi, Gustavo Jardim Volpe, Bruno Belmonte Martinelli Gomes, Maria Aparecida Alves Leite Dos Santos Almeida, Glenda Renata de Moraes, Simone Kashima, Marcos Carvalho Borges.

**Formal analysis:** Natasha Nicos Ferreira, Pedro Manoel Marques Garibaldi, Gustavo Jardim Volpe, Bruno Belmonte Martinelli Gomes, Marcos Carvalho Borges.

**Resources:** Maria Aparecida Alves Leite Dos Santos Almeida, Glenda Renata de Moraes, Dimas Tadeu Covas, Simone Kashima, Rodrigo Tocantins Calado.

**Supervision:** Gustavo Jardim Volpe, Marcos Carvalho Borges.

**Writing – original draft:** Natasha Nicos Ferreira.

**Writing – review & editing:** Natasha Nicos Ferreira, Pedro Manoel Marques Garibaldi, Gustavo Jardim Volpe, Bruno Belmonte Martinelli Gomes, Maira Nilson Benatti, Maria Aparecida Alves Leite Dos Santos Almeida, Rodrigo Tocantins Calado, Benedito Antonio Lopes Fonseca, Marcos Carvalho Borges.

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
