## [Decision Letter · Decision Letter 0]

12 Aug 2025

PGPH-D-24-03019

Clinical, epidemiological, and socioeconomic characteristics associated with SARS-CoV-2 reinfection: an observational study.

Dear Dr. Ferreira,

Thank you for submitting your manuscript to PLOS Global Public Health. After careful consideration, we feel that it has merit but does not fully meet PLOS Global Public Health’s publication criteria as it currently stands. Therefore, we invite you to submit a revised version of the manuscript that addresses the points raised during the review process.

We look forward to receiving your revised manuscript.

Kind regards,

Rajiv Sarkar

Academic Editor

Journal Requirements:

2. Please note that your Data Availability Statement is currently missing the repository name. If your manuscript is accepted for publication, you will be asked to provide these details on a very short timeline. We therefore suggest that you provide this information now, though we will not hold up the peer review process if you are unable.

Additional Editor Comments:

In addition to responding to the comments from the reviewers, please also address the following comments:

(1) Please change 'descriptive and observational cohort' to 'cohort' as cohorts are observational studies by design. Also, kindly highlight how reinfections were identified - was it solely based on medical records or was any active follow-up conducted? If later, what was the frequency & duration of follow-up, & how many participants could the researchers contact during the follow-up period (i.e., was there any loss to follow-up)? If reinfections were identified through patient medical records, is there a possibility of reporting bias (i.e., some reinfections were missed as participants did not report to the hospital/undergo testing)?

(2) Please clarify how the epidemiological risk-factor data was collected. Was it through a questionnaire or was it based on patient medical records? If through a questionnaire, was the questionnaire self-reported or interviewer administered? How was the questionnaire developed & validated?

Also, when was the risk-factor data obtained? Was it at the time of initial COVID-19 diagnosis or was it at the time of diagnosis of reinfection? If the risk-factor data was obtained at the time of initial infection, is there any chance of misclassification bias (esp. for the preventive measures as the practices such as mask wearing or handwashing may change over time)?

(3) From the description it seems that the 'first documented infection' (n=11,129) was actually the number of participants with a single documented COVID-19 infection. Similarly, the 'second infection' (n=890) were patients with two documented infections & 'third-time infection' (n=32) were patients with three documented infections. If so, kindly mention it explicitly to avoid ambiguity.

(4) It will be interesting to examine if certain viral variants were more prone to reinfection than others. The authors seem to have the requisite data for this analysis.

Reviewers' comments:

Reviewer's Responses to Questions

**Comments to the Author**

1. Does this manuscript meet PLOS Global Public Health’s publication criteria?

Reviewer #1: Yes

Reviewer #2: Yes

2. Has the statistical analysis been performed appropriately and rigorously?

Reviewer #1: Yes

Reviewer #2: Yes

3. Have the authors made all data underlying the findings in their manuscript fully available (please refer to the Data Availability Statement at the start of the manuscript PDF file)?

Reviewer #1: No

Reviewer #2: Yes

4. Is the manuscript presented in an intelligible fashion and written in standard English?

Reviewer #1: Yes

Reviewer #2: Yes

Reviewer #1: Abstract: Make sure there are no errors with presentation of results. Ideal scenario use 95% Confidence interval instead of p-values. (Line 40: 95% p<0.001, and Line 44).

Results: Slight improvements needed in presentation of results in tables. And additional description for preference of Poisson regression model.

Reviewer #2: The article is relevant and provides real-world information that allows for considerations in health surveillance for decision-making regarding population protection, and therefore has a global public health aspect.

However, part of the information provided and final considerations could be presented in the introduction to clarify to the reader that it is a specific aspect in a municipality with an implemented health surveillance system, but which has particular characteristics.

It is suggested to change the title, given that clinical and epidemiological aspects were not deeply explored, but rather demographic and socioeconomic aspects as characteristics associated with SARS-CoV-2 re-infection.

Characterisation of the municipality of Serrana in the interior of the State of São Paulo, Brazil

Given that demographic characteristics were mentioned, a greater characterisation of the municipality of Serrana is warranted, being a small town in the interior of a state where rural activities seem to be relevant, as well as its proximity to a larger hub municipality. Thus, a more adequate characterisation would also involve the flows related to the relationships between these municipalities.

Improve Keywords: The keywords do not facilitate the search for the real theme of the article, such as demography, pandemic, health surveillance, and vaccination.

Line 70: The reference presented in line 70 regarding surveillance in the municipality of Serrana, written by the same primary author, does not show the DOI - https://doi.org/10.1016/j.puhip.2022.100301, which would facilitate finding the article that, as verified, is closely related to the submitted manuscript.

In line 147, the wording should be reconsidered, given that both continuous and discrete variables have a quantitative aspect.

Lines 149 and 150: Clarification is suggested regarding the term qualitative, given that, although it is a qualifier, the reader may have doubts whether the proposal will bring qualitative information related to this type of research, which is not the case. It is a study with a quantitative and descriptive aspect.

Furthermore, the term gender was repeatedly used in the text; it would be appropriate to consider using the term sex, given that issues related to gender are not addressed, but rather the characterisation between male and female.

Given the different statistical methods used, it is suggested to create a table that presents method versus purpose of use.

Another important aspect is to make it clear that the waves and months referred to relate to the identification of variants according to the surveillance system implemented in the municipality used as a reference, and not what was identified in the international scenario.

Given the same ethics committee approval code between the presented manuscript and the work pointed out by the main author in the bibliography, clarification is needed regarding the ethics committee's affiliated institution. Clearly state the committee's affiliation.

In line 207, the situation where the use of the term sex and not gender is suggested is well illustrated, given that it refers to the differentiation between male and female.

Line: 145: In terms of wording and for the terminology to be well framed in genomic characterisations, it should be noted that BA.2 refers to a sub-lineage of the Omicron variant.

Insert DOI in the main author's reference

Ferreira NN, Garibaldi 358 PMM, Moraes GR, Moura JC, Klein TM, Machado LE, et al.

359 The impact of an enhanced health surveillance system for COVID-19 management

360 in Serrana, Brazil. Public Health in Practice. 2022;4.

**Do you want your identity to be public for this peer review?** For information about this choice, including consent withdrawal, please see our Privacy Policy

Reviewer #1: No

Reviewer #2: No

---

## [Decision Letter · Decision Letter 1]

21 Nov 2025

PGPH-D-24-03019R1

Demographic and socioeconomic characteristics associated with SARS-CoV-2 reinfection: an observational study.

Dear Dr. Ferreira,

Thank you for submitting your manuscript to PLOS Global Public Health. After careful consideration, we feel that it has merit but does not fully meet PLOS Global Public Health’s publication criteria as it currently stands. Therefore, we invite you to submit a revised version of the manuscript that addresses the points raised during the review process.

We look forward to receiving your revised manuscript.

Kind regards,

Rajiv Sarkar

Academic Editor

Journal Requirements:

Additional Editor Comments (if provided):

In addition to responding to the comments from Reviewer # 1, please also address the following:

(1) From the description in lines 219-222, there were 922 documented reinfections (890  second infections and 32 third infections), which would result in a reinfection rate of 7.6% (922/12051) and not 8.28%, as mentioned. Kindly cross-check.

(2) For the details on the severity of SARS-CoV-2 reinfections and viral variants (lines 270-294), kindly specify how many observations were available for the analysis as, from Table 3, it seems that the data was available for a very small subset of cases (330 with single infection and 15 with two or more infections, i.e., a total of 345). Also, the numbers in Table 3 does not match with that presented in lines 283-286 on the number of reinfections by viral variant, which adds up to 374 reinfections (20 + 43 + 301 + 10). Why this discrepancy?

(3) How was the incidence of reinfections by viral variants (lines 289-292) calculated? From the data presented in lines 283-286, the highest percentage of reinfections (13.7%) were observed in cases with the Omicron BA.2 sub-lineage, although the incidence of reinfection in that variant was lower than the Delta or the Omicron variant, both of which had lower precentage of reinfections (2.3% and 7.9% respectively). Kindly cross-check.

(4) Please round the percentages to one decimal place. The RRs and 95% CIs can be rounded to two decimal places, and p-values to three decimal places.

(5) Please thoroughly proof-read the manuscript as there are a few typos.

Reviewers' comments:

Reviewer's Responses to Questions

**Comments to the Author**

Reviewer #1: All comments have been addressed

Reviewer #2: All comments have been addressed

publication criteria?

Reviewer #1: Yes

Reviewer #2: Yes

3. Has the statistical analysis been performed appropriately and rigorously?

Reviewer #1: Yes

Reviewer #2: Yes

4. Have the authors made all data underlying the findings in their manuscript fully available (please refer to the Data Availability Statement at the start of the manuscript PDF file)?

Reviewer #1: Yes

Reviewer #2: Yes

5. Is the manuscript presented in an intelligible fashion and written in standard English?

Reviewer #1: Yes

Reviewer #2: Yes

Reviewer #1: 1. Authors mentioned that females were at greater risk than males for reinfection. In the discussion there are reasons given as to why that is the case. Sex is not a modifiable risk factor, however activities potentially done by women and not men may be the reason for reinfections. These could be the areas of public health interventions.

2. Line 43-46 prevalence was presented as a relative risk. Could this be a typo?

3. Line 186-188, since this was a cohort study during COVID, Odds ratios would not have been appropriate based on the study design as well as the fact that COVID-19 infection and reinfection were not rare outcomes during the period of investigation. Perhaps it is not necessary to justify RR over OR.

4. If you can improve the discussion by being more direct about why risk factors were seen as such. AT the moment it looks like a comparison of your results and those of other studies.

Reviewer #2: (No Response)

**Do you want your identity to be public for this peer review?** For information about this choice, including consent withdrawal, please see our Privacy Policy

Reviewer #1: No

Reviewer #2: No

---

## [Decision Letter · Decision Letter 2]

26 Jan 2026

PGPH-D-24-03019R2

Demographic and socioeconomic characteristics associated with SARS-CoV-2 reinfection: an observational study.

Dear Dr. Ferreira,

Thank you for submitting your manuscript to PLOS Global Public Health. After careful consideration, we feel that it has merit but does not fully meet PLOS Global Public Health’s publication criteria as it currently stands. Therefore, we invite you to submit a revised version of the manuscript that addresses the points raised during the review process.

We look forward to receiving your revised manuscript.

Kind regards,

Rajiv Sarkar

Academic Editor

Journal Requirements:

Additional Editor Comments (if provided):

In addition the addressing the reviewer's comments below, please also address the following:

Tables 2 and S2: Please add a footnote describing how the adjusted RRs were calculated. Additionally, did you consider conducting a multivariable analysis to identify the key risk factors for reinfection, while adjusting for potential confounders?Table 3: Please include a footnote specifying the statistical tests used to calculate the p-values. Given the small number of cases with multiple (two or three) infections, were the assumptions underlying the statistical tests met?Table S1: Please replace the term “Gender” with “Sex,” to conform with the terminology used in Tables 1–3. Additionally, categorize participants as “Females” and “Males” rather than “Feminine” and “Masculine”.Severity of SARS-CoV-2 reinfections: Did the severity of reinfections differ by the viral variant, particularly considering the fact that the Delta variant was more pathogenic than Alpha or Omicron variants?

Reviewers' comments:

Reviewer's Responses to Questions

**Comments to the Author**

Reviewer #1: All comments have been addressed

publication criteria?

Reviewer #1: Yes

3. Has the statistical analysis been performed appropriately and rigorously?

Reviewer #1: Yes

4. Have the authors made all data underlying the findings in their manuscript fully available (please refer to the Data Availability Statement at the start of the manuscript PDF file)?

Reviewer #1: Yes

5. Is the manuscript presented in an intelligible fashion and written in standard English?

Reviewer #1: Yes

Reviewer #1: The revision 2 is a marked improvement from revision 1. A couple of minor issues issues:

Line 250-252: Is this the correct way of interpreting. if 6-9 minimum wages is the baseline, then compared to 3 minimum wages the risk is lower? can you double check on this and other interpretations for the same variable.

In the excel file, it looks like some under variable number of people on household it looks like some people's names are in that column. I uploaded the csv file into STATA and tabulated the variable 'peoplelivinginthesamehouse'. Please double check on this and potentially reanalyze your data.

Line 84-85: Were the coordinates necessary? Unless you wanted to do a GIS type of analysis?

**Do you want your identity to be public for this peer review?**

Privacy Policy

Reviewer #1: No

---

## [Decision Letter · Decision Letter 3]

15 Feb 2026

PGPH-D-24-03019R3

Demographic and socioeconomic characteristics associated with SARS-CoV-2 reinfection: an observational study.

Dear Dr. Ferreira,

Thank you for submitting your manuscript to PLOS Global Public Health. After careful consideration, we feel that it has merit but does not fully meet PLOS Global Public Health’s publication criteria as it currently stands. Therefore, we invite you to submit a revised version of the manuscript that addresses the points raised during the review process.

We look forward to receiving your revised manuscript.

Kind regards,

Rajiv Sarkar

Academic Editor

Journal Requirements:

Additional Editor Comments (if provided):

Tables 1 and 2: Monthly family income – please mention the $-value for each category in parenthesis or as footnote, similar to what has been presented in S1 Table.

Discussion, lines 365-366: ‘Then, the vaccination scheme might have influenced the clinical outcomes of patients with COVID-19 reinfections.’ Should lines 382-385 not immediately follow this sentence, with an explanation of how the COVID-19 reinfections may have been influenced by the vaccination scheme and previous infections?

Figure 2: Please consider adding a second y-axis for 2^nd^ and 3^rd^ infections as their numbers are too small compared to the number of 1^st^ infections.

Reviewers' comments:

Reviewer's Responses to Questions

**Comments to the Author**

Reviewer #1: All comments have been addressed

publication criteria?

Reviewer #1: Yes

3. Has the statistical analysis been performed appropriately and rigorously?

Reviewer #1: Yes

4. Have the authors made all data underlying the findings in their manuscript fully available (please refer to the Data Availability Statement at the start of the manuscript PDF file)?

Reviewer #1: Yes

5. Is the manuscript presented in an intelligible fashion and written in standard English?

Reviewer #1: Yes

Reviewer #1: all comments have been addressed

**Do you want your identity to be public for this peer review?** For information about this choice, including consent withdrawal, please see our Privacy Policy

Reviewer #1: No

---

## [Editor Report · Decision Letter 4]

24 Feb 2026

Demographic and socioeconomic characteristics associated with SARS-CoV-2 reinfection: an observational study.

PGPH-D-24-03019R4

Dear Ms Ferreira,

We are pleased to inform you that your manuscript 'Demographic and socioeconomic characteristics associated with SARS-CoV-2 reinfection: an observational study.' has been provisionally accepted for publication in PLOS Global Public Health.

Best regards,

Rajiv Sarkar

Academic Editor
